# Child maltreatment, early life socioeconomic disadvantage and all-cause mortality in mid-adulthood: findings from a prospective British birth cohort

Nina T Rogers ,[1] Christine Power,[2] Snehal M Pinto Pereira [1]

[1]UCL Division of Surgery & Interventional Science, University College London, London, UK
[2]UCL Great Ormond Street Institute of Child Health, London, UK

**Correspondence to**
Dr Snehal M Pinto Pereira; snehal.pereira@ucl.ac.uk

## ABSTRACT

**Objectives** Early-life adversities (ELAs) such as child maltreatment (neglect and abuse) and socioeconomic disadvantage have been associated with adult mortality. However, evidence is sparse for specific types of ELA. We aimed to establish whether specific ELAs (ie, different types of child maltreatment and socioeconomic disadvantage) were associated independently with all-cause mortality in mid-adulthood and to examine potential intermediary pathways.

**Design** Prospective cohort study.

**Setting** 1958 British birth cohort: a longitudinal, population-based sample of individuals born in Great Britain during a single week in March 1958.

**Participants** 9310 males and females with data on child maltreatment and mortality (44/45–58 years).

**Outcome measures** Mortality follow-up from 2002/2003 to 2016 when participants were aged 44/45–58 years. Death was ascertained via the NHS Central Register (N=296) or cohort maintenance activities (N=16).

**Results** Prevalence of ELAs ranged from 1.6% (sexual abuse) to 11% (psychological abuse). Several, but not all, ELAs were associated with increased risk of premature death, independent of covariates and other adversities; adjusted HRs were 2.64 (95% CI 1.52 to 4.59) for sexual abuse, 1.93 (95% CI 1.45 to 2.58) for socioeconomic disadvantage, 1.73 (95% CI 1.11 to 2.71) for physical abuse and 1.43 (95% CI 1.03 to 1.98) for neglect. After adjustment for covariates and other adversities, no associations with mortality were observed for psychological and witnessing abuse. Regarding potential intermediaries (including adult socioeconomic factors, behaviours, adiposity, mental health and cardiometabolic markers), most associations attenuated after accounting for adult health behaviours (particularly smoking). In addition, early-life socioeconomic disadvantage and neglect associations attenuated after accounting for adult socioeconomic factors. The association for sexual abuse and premature mortality was largely unaffected by potential intermediaries.

**Conclusions** Associations with premature mortality varied by type of ELA: associations for sexual and physical abuse, neglect and socioeconomic disadvantage were independent of each other. Different types of ELAs could influence premature mortality via different pathways; this requires further research.

---

### Strengths and limitations of this study

► Data were from a large population-based cohort followed from birth, which allowed prospective ascertainment of child neglect, early-life socioeconomic disadvantage, important covariates and potential intermediary factors.

► Information on different types of child abuse was reported retrospectively at 45 years.

► Data on cause-specific mortality were not available; however, mortality data were collected over 14 years.

## BACKGROUND

Early-life adversities (ELAs) such as child maltreatment (CM) (neglect and abuse) and socioeconomic disadvantage are major public health issues.[1 2] These adversities are not uncommon, for example, in the UK, approximately 9% of children and 22% of adolescents are estimated to experience neglect and/or physical, psychological or sexual abuse,[3] and approximately 4.6 million children live in poverty.[4] Although a growing literature links CM to several poor health outcomes[5–7] in childhood through to older ages, evidence on links to mortality in adulthood is based primarily on adversity scores,[8–12] which include other experiences such as living in a household with someone who has previously been imprisoned. While a more extensive literature including systemic reviews[2] demonstrates associations for early-life socioeconomic disadvantage, few studies[9] consider both CM and early-life socioeconomic disadvantage simultaneously in relation to mortality in adulthood. This omission is important because, although these ELAs are related, they represent distinct concepts[13] with potential differences in mortality risk. Clarifying the extent to which CM associates

with later mortality independent of childhood socioeconomic background (and vice versa) would inform understanding of the role of different types of early-life exposures.

Moreover, it is possible that associations vary for *specific* types of CM and mortality in adulthood, given the differences reported for several outcomes in adulthood, including socioeconomic disadvantage,[14 15] mental[16–18] and physical[6] health. Variation in associations with adult mortality cannot be determined from the sparse literature available to date for specific types of CM. One US study that combined multiple CMs together, found no risk of premature mortality although follow-up was limited to young adulthood.[19] In a second US study, associations were found in women but not men for physical and psychological abuse with all-cause mortality over 20 years of follow-up from ages 25 years to 74 years at baseline; information on sexual abuse and neglect was unavailable.[9] Thus, previous studies have investigated CM as a combined score of different types[9 19] or a limited number of types examined separately.[9] With such limitations of research conducted to date, possible differential associations for specific types of CM (ie, their independence from each other as well as from early-life socioeconomic disadvantage) on mortality in adulthood are not well understood.

With respect to potential pathways from ELAs to adult mortality, it is well established that, for example, CM is associated with detrimental factors in adulthood, including socioeconomic circumstances,[14 15] risky health behaviours (eg, smoking, drug misuse, problem drinking),[17 20 21] obesity,[6] poor mental[16–18] and physical health.[22] In turn, these factors are linked to mortality.[23–25] Yet existing literature examining such intermediaries is limited, particularly in relation to CM. Understanding pathways through which specific types of ELAs link to mortality in mid-adulthood is important for developing appropriate interventions that aim to reduce inequalities in mortality.

Given current knowledge gaps, we aimed to establish in a general population sample followed from birth: (1) the extent to which CM and early-life socioeconomic disadvantage are associated independently with premature mortality in midlife (from 44/45 years to 58 years) and whether associations vary by type of ELA and (2) whether associations are explained by potential intermediaries including adult socioeconomic, behavioural, adiposity, mental health and cardiometabolic status.

## METHODS

The 1958 British birth cohort consists of over 17 000 participants followed-up since birth during 1 week of March 1958.[26] Respondents in mid-adulthood are broadly representative of the surviving cohort.[27] Of 11 971 invited at 44/45 years, 9310 completed at least one question on CM and had information on mortality (44/45–58 years) (see online supplemental figure 1).

### Early-life adversities

Socioeconomic disadvantage was identified from prospectively recorded information on father's occupation at the participant's birth. Those with a father in an unskilled manual occupation were classified as disadvantaged. Child neglect was identified from prospectively collected information at 7 years and 11 years from interview questions to the child's mother and teacher. Eleven indicators of neglect were selected to represent conventional definitions and were summed to create a score (range 0–11). A score ≥3 yielded a prevalence estimate in line with other UK estimates,[1 3] and was used here to define child neglect. Childhood (0–16 years) physical, psychological, witnessing and sexual abuse was reported retrospectively at 44/45 years using a confidential computer-assisted data-entry questionnaire. Child neglect and abuse measures have been used in several previous studies that, reassuringly, provide extensive evidence of construct validity.[28] Details of all ELAs are given in table 1.

### All-cause mortality

Information on deaths between 2002/2003 and end of 2016 was ascertained from a variety of sources, mostly (N=296) through receipt of death certificates (including date of death) from the National Health Service Central Register. Information from relatives or close friends during survey activities/cohort maintenance allowed identification of 16 further deaths (details in table 2 footnotes). Cause-specific data were not available.

### Covariates

Covariates were selected a priori. All were prospectively recorded, including maternal age at birth, birth weight (adjusted for gestational age), birth order and physical or cognitive impairment at age of 7 years. Additional covariates for CM analyses included social class at birth and household factors (amenities, tenure and crowding) at 7 years (details in figure 1 footnotes). Birth weight was ascertained from clinical records; parents reported all other factors.

### Potential mid-adult intermediary factors

Potential mid-adult intermediary factors were selected based on established associations with both ELAs and premature mortality. Details of included factors are given in online supplemental table S1, that is, for adult (1) socioeconomic factors: age 33 years social class and educational qualifications, (2) behavioural factors: age 42 years smoking, age 45 years problem drinking and age 42 years illegal drugs use in the last 12 months; (3) adiposity: age 45 years obesity and waist-hip ratio; (4) age 42 years mental health and (5) cardiometabolic factors: age 45 years glycated haemoglobin, triglycerides and low-density lipoprotein cholesterol (LDL-c), all adjusted for medications. Cardiometabolic factors, height, weight, hip and waist were measured by trained professionals; other factors were self-reported. Most considered intermediaries were associated with mortality in this cohort (online supplemental table S2).

**Table 1**  Definition of early-life adversities (child maltreatment and early-life socioeconomic disadvantage) and representative variables from the 1958 British birth cohort

| | Definition* | 1958 cohort variables | Age of ascertainment (method)† |
|---|---|---|---|
| **Prospective report, birth to 11 years** | | | |
| Socioeconomic disadvantage (birth) | | Based on father's occupation at birth,‡ using the Registrar General's Classification. Fathers with an unskilled manual occupation or households with no male head were classified as disadvantaged. | Birth (P) |
| Neglect§ (7 years and 11 years) | Failure to meet a child's basic physical, emotional, medical/dental or education need; failure to provide adequate nutrition, hygiene, or shelter; or failure to ensure a child's safety | Child looks undernourished, scruffy or dirty | 7 and 11 years (T) |
| | | Mother never/hardly ever takes child out¶ | 7 and 11 years (P) |
| | | Father never/hardly ever takes child out¶ | 7 and 11 years (P) |
| | | Mother shows little/no interest in child's educational progress | 7 and 11 years (T) |
| | | Father shows little/no interest in child's educational progress | 7 and 11 years (T) |
| | | Mother and Father never/hardly ever read to, or reads with child | 7 years (P) |
| **Retrospective report at 44/45 years** | | | |
| Physical abuse (0–16 years) | Intentional use of physical force or implements against a child that results in, or has the potential to result in, physical injury. | I was physically abused by a parent—punched, kicked or hit or beaten with an object, or needed medical treatment | 45 years (S) |
| Psychological abuse** (0–16 years) | Intentional behaviour that conveys to a child that s/he is worthless, flawed, unloved, unwanted, endangered or valued only in meeting another's needs. UK definition†† includes harmful (unintentional) parent–child interactions: 'the persistent emotional maltreatment of a child such as to cause severe and persistent adverse effects on the child's emotional development' | I was verbally abused by a parent (or parent-figure) | 45 years (S) |
| | | I suffered humiliation, ridicule, bullying or mental cruelty from a parent (or parent-figure) | |
| | | Mother (or mother-figure) and father (or father-figure) were not at all affectionate | |
| Witnessing abuse (0–16 years) | Any incident of threatening behaviour, violence or abuse (psychological, physical, sexual, financial or emotional) between intimate partners or adult family members, irrespective of sex or sexuality | I witnessed physical or sexual abuse of others in my family | 45 years (S) |
| Sexual abuse (0–16 years) | Any completed or attempted sexual act, sexual contact or non-contact sexual interaction with a child by a caregiver | I was sexually abused by a parent (or parent-figure) | 45 years (S) |

*Gilbert et al.[1]
†(S): self-report; (T): teacher-report; (P): parent-report.
‡Socioeconomic position was classified as missing for fathers' who were unemployed or sick.
§Questions relating to child neglect at 7 years and 11 years were answered by the child's teacher and mother (or father if the mother was unavailable). The 11 neglect indicators were summed to create a score (range 0–11); those scoring > 3 were classified as neglected (see text for further details).
¶For example, walks, outings, picnics, visits, shopping.
**In the 1958 cohort psychological abuse was defined as experiencing at least one of the three listed variables.
††Department for Education. Working together to safeguard children. Her Majesty's Government, 2006.

## Statistical analysis

Cox proportional hazard models were used to estimate hazard ratios (HR) and 95% CIs for associations between each type of ELA and mortality. Survival time included the time from completion of the 44/45 years questionnaire to the date of death, censoring (last date of contact) or the end of the study period (December 2016), whichever came first. Schoenfeld residuals were examined to test the assumption of proportional hazards for covariates and potential intermediaries; none violated the assumption ($p\geq0.12$).

We examined associations between each type of ELA and mortality in separate analyses for men and women and also tested whether associations differed using an interaction term (ie, type of ELA and mortality by sex) in analyses of both sexes combined. There was little evidence of effect modification ($p_{sex*ELA}\geq0.28$ and online supplemental table S3), hence in a first level of analyses we adjusted for sex (model

**Table 2** Prevalence of early-life adversities and mortality* in the 1958 British birth cohort

| Early-life adversity | Population sample N† | Total cases N (%) | Males (%) | Females (%) | Deaths N (%) |
|---|---|---|---|---|---|
| Socioeconomic disadvantage | 9033 | 925 (10.2) | 9.6 | 10.8 | 61 (6.6) |
| Neglect‡ | 8460 | 878 (10.4) | 11.1 | 9.7 | 49 (5.6) |
| Physical abuse | 9308 | 562 (6.0) | 5.9 | 6.1 | 40 (7.1) |
| Psychological abuse | 9310 | 1000 (10.7) | 8.9 | 12.6 | 50 (5.0) |
| Witnessing abuse | 9308 | 559 (6.0) | 4.4 | 7.6 | 33 (5.9) |
| Sexual abuse | 9308 | 149 (1.6) | 0.5 | 2.7 | 17 (11.4) |
| Deaths 44/45–58 years* | 9310 | 312 (3.4) | 3.7 | 3.0 | |

*Date of death was ascertained through receipt of death certificates to the Centre for Longitudinal Studies from the National Health Service Central Register (N=296), that is, data missing for 16 individuals (see: National Child Development Study Deaths Dataset, 1958–2016 UK Data Service for details). Using survey/cohort maintenance data we determined if the deceased died between 45–50 years (N=7), 50–55 years (N=5) and 55–58 years (N=4). Date of death was estimated as the midpoint between these ages.
†N varies due to missing data.
‡Those with complete data on 6 or more of 11 neglect items (as detailed in Power et al).[28]

1). Second, to assess whether associations were independent of other early-life factors, we additionally adjusted for covariates listed above (model 2). Third, because different types of ELAs often cluster,[29] we assessed two-way correlations between examined ELAs (online supplemental table S4). Most ELAs were weakly or only modestly correlated (phi coefficient ≤0.50). Therefore, in model 3, we adjusted for all types of ELA simultaneously. For associations that remained

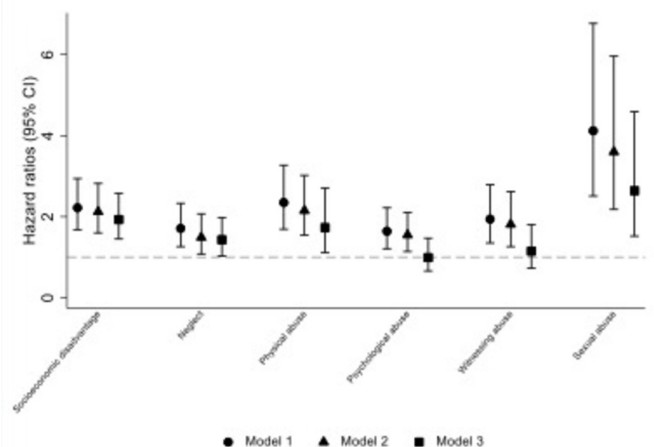

**Figure 1** Hazard ratios (95% confidence intervals) for early-life adversities in relation to all-cause mortality in 1958 birth cohort participants aged 44/45 years to 58 years (N=9310). Model 1: adjusted for sex only; model 2: additionally adjusted for maternal age at birth, birth weight (adjusted for gestational age), birth order and physical or cognitive impairment (yes/no) at age of 7 years. For associations with neglect, physical, psychological, witnessing and sexual abuse (but not for early-life disadvantage) models additionally adjusted for socioeconomic factors: social class at birth (or if missing, at age 7 years), age 7 years household amenities (sharing or lack a bathroom, lavatory or hot water), age 7 years housing tenure (owner/occupier, renter or other) and age 7 years household crowding (1+ person/room); model 3: model 2 plus simultaneous adjustments for all other early-life adversities.

in model 3, we assessed the role of potential intermediaries (socioeconomic, behavioural, adiposity, mental health and cardio-metabolic factors) in explaining ELA—mortality associations, by further adjusting model 3 for each potential intermediary (in groups as well as for each factor separately).

In sensitivity analyses, we checked whether restricting the sample to those completing the CM questions at 44/45 years affected results, by repeating analyses using the larger sample available for child neglect and socioeconomic disadvantage (N=15 092). Survival time included the time from completion of the age of 11 years survey to the date of death, censoring or the end of the study period, whichever came first. As an additional check on the independence of associations for different types of ELAs from model 3, we examined associations with mortality for groups with only one specific type of ELA vs no ELA.

Missing data ranged from 0.02% (physical, sexual and witnessing abuse) to 21% (LDL-c) (online supplemental table S5). Data loss was minimised, by imputing missing data on all substantive model variables (ie, all variables included in models 1, 2 and/or 3) using multiple imputation chained equations. Imputation models included all substantive model variables and main predictors of missingness.[27] Regression analyses were run across 20 imputed datasets and overall estimates obtained. Imputed results were similar to those obtained using observed values (online supplemental table S6); the former are presented.

### Patient and public involvement
Patients and the public were not involved in the design of the study, or in the interpretation or writing up of the manuscript.

### RESULTS
The prevalence of ELAs varied from 1.6% (sexual abuse) to 11% (psychological abuse) with 10% classified as socioeconomically disadvantaged in early life (table 2). The majority of participants reported no ELA (71%) with

**Table 3** Early-life adversities and risk of all-cause mortality (HR (95% CI)) in adults (44/45–58 years) adjusted for mid-adult (1) socioeconomic, (2) behavioural, (3) adiposity, (4) mental health and (5) cardiometabolic factors* (N=9310)

| | Socioeconomic disadvantage | Neglect | Physical abuse | Sexual abuse |
|---|---|---|---|---|
| Model 3 | 1.93 (1.45 to 2.58) | 1.43 (1.03 to 1.98) | 1.73 (1.11 to 2.71) | 2.64 (1.52 to 4.59) |
| +mid-adult socioeconomic factors | 1.82 (1.36 to 2.43) | 1.28 (0.91 to 1.79) | 1.74 (1.11 to 2.73) | 2.54 (1.46 to 4.42) |
| +mid-adult behavioural factors | 1.75 (1.31 to 2.34) | 1.32 (0.95 to 1.83) | 1.50 (0.96 to 2.34) | 2.43 (1.40 to 4.23) |
| +mid-adult adiposity | 1.90 (1.42 to 2.53) | 1.39 (1.00 to 1.93) | 1.75 (1.12 to 2.73) | 2.71 (1.56 to 4.73) |
| +mid-adult mental health | 1.91 (1.43 to 2.55) | 1.37 (0.99 to 1.91) | 1.70 (1.09 to 2.64) | 2.54 (1.46 to 4.42) |
| +mid-adult cardio-metabolic factors | 1.89 (1.42 to 2.53) | 1.39 (1.00 to 1.93) | 1.65 (1.05 to 2.57) | 2.71 (1.56 to 4.72) |

Model 3 (adjustments shown in figure 1 footnotes).
*Models were adjusted for each intermediary group of factors separately (not simultaneously). See details of intermediary factors in online supplemental table S1.

19% reporting one and 10% reporting two or more types of ELA. Between 44/45 years and 58 years, 3.4% of the sample died (N=312).

All types of ELA were associated with risk of death (44/45–58 years) after controlling for covariates (model 2; figure 1 and online supplemental table S6), for example HR for neglect was 1.49 (95% CI 1.08 to 2.07) and for physical abuse was 2.15 (95% CI 1.54 to 3.02). In models simultaneously adjusted for all other types of ELA (model 3) associations remained for all except psychological and witnessing abuse, namely for neglect (HR 1.43 (95% CI 1.03 to 1.98)), physical abuse (HR:1.73 (95% CI 1.11 to 2.71)), sexual abuse (HR 2.64 (95% CI 1.52 to 4.59)) and socioeconomic disadvantage (HR 1.93 (95% CI 1.45 to 2.58)). The reduction in HRs between models 2 and 3 was seen consistently for all ELAs, although modest in some instances, for example, for early-life socioeconomic disadvantage the HR reduced from 2.12 (95% CI 1.60 to 2.82) to 1.93 (95% CI 1.45 to 2.58) after adjusting for all CMs.

In regard to potential intermediaries, associations between ELAs and death in mid-adulthood were largely unaffected by adjustment for the range of factors examined (table 3). However, most associations attenuated after adjustment for adult health behaviours, for example, HRs for physical abuse attenuated from 1.73 (95% CI 1.11 to 2.71) to 1.50 (95% CI 0.96 to 2.34). Separate adjustment for each health behaviour in turn showed a predominant attenuating effect of smoking (online supplemental table S7). Additionally, associations for neglect and early-life socioeconomic disadvantage attenuated after controlling for adult socioeconomic factors. For sexual abuse and early-life socioeconomic disadvantage reductions in the strong associations with mortality in mid-adulthood were negligible after accounting for intermediaries.

In sensitivity analysis using the larger sample available for child neglect and socioeconomic disadvantage (N=15 092), main findings were largely unaltered to those presented in figure 1 (online supplemental table S8). Checks on the independence of associations performed for groups with only one specific type of ELA (vs no ELA) showed broadly similar mortality associations (although with wider CIs) to main results in figure 1 model 3 (online supplemental table S9).

## DISCUSSION

In this large population-based study on different types of ELA and mortality in mid-adulthood, we showed several important findings. First, some ELAs, but not all, were associated with higher risk of premature mortality in mid-adulthood. That is, findings varied by type of adversity. Child sexual abuse was strongly associated with mortality with a 2.6 times higher risk of premature death in mid-adulthood, although it was the least prevalent adversity. For early-life socioeconomic disadvantage, experienced by 10% of the population, there was an approximate doubling in risk of premature mortality. For physical abuse and neglect the estimated elevated risk of death was more modest (73% and 43% higher, respectively), whereas no associations were observed for psychological and witnessing abuse. Second, observed associations were independent of potential confounding factors and the other adversities examined. Importantly, the specific CM associations were mostly robust when accounting for early-life socioeconomic disadvantage and vice versa. Third in relation to potential intermediaries, associations for all types of ELA attenuated after controlling for adult health behaviours, in particular smoking. But, in some instances this attenuation was minor, such that for sexual abuse the association was largely unaltered. Associations for early-life socioeconomic disadvantage and neglect were also attenuated by adult socioeconomic factors. Other examined intermediaries including cardiometabolic markers did little to explain observed associations between specific CMs and mortality in mid-adulthood or for early-life socioeconomic disadvantage.

Our study has several notable strengths. The range of data available on different types of ELA facilitated simultaneous analysis to inform on their independent effects. This is essential for investigating distinct effects of CM on mortality, that is, separate from those of socioeconomic background and also, in regard to specific types of CMs.

Inclusion of child neglect is particularly important given that it is often ignored in research on CM.[30] A follow-up of approximately 14 years is a further study strength, as is use of linked mortality data, which is independent of ongoing study participation. Alongside the 14 years mortality follow-up there are advantages of using a single-age sample in reducing the range of possible causes of premature death and related underlying pathways. However, study limitations are acknowledged. Ascertainment of childhood maltreatment is not straightforward, with limitations noted for all methods,[1] including those used here. While child neglect indicators were measured prospectively and included many aspects of the conventional definition (eg, failure to ensure a child's basic physical, emotional and educational needs), there were some omissions (eg, failure to ensure a child's safety) and neglect after age 11 years may be missed. However, our measure uses information from different sources (parents and teachers) and at two ages (7 years and 11 years) which may reduce misclassification and rather than relying on individual items, we used a composite score. Abuse by a parent (up to age 16 years) was reported retrospectively and does not include abuse by others possibly leading to an underestimate of prevalence. Nonetheless, prevalence estimates of CM were generally in keeping with previous approximates for the UK.[1 3] An exception is child sexual abuse where prevalence is low and estimates may be underpowered. Thus, we have used both prospective (neglect and early-life socioeconomic disadvantage) and retrospective (abuse) measures and we acknowledge that these may identify different groups of individuals.[31] However, it is reassuring that a broad range of studies based on our measures of child neglect and abuse provide extensive evidence of construct validity.[28] As with any long-term study, selection bias needs to be considered: by 45 years, when information was collected on child abuse, not all in the cohort had survived (6.7% had died); however, over half of these deaths had occurred before age of 7 years (mostly in the first months of life).[27] Selection bias may affect findings reported here, but only if patterns of association with mortality differ in the surviving and deceased populations. Relatedly, sensitivity analysis for child neglect and socioeconomic disadvantage in a larger sample with longer follow-up from age 11 years suggests that study results are robust. The analytical approach used allows insights into possible mechanisms underlying ELA associations with premature mortality, but it is not possible to determine the most important intermediaries without considering confounding between mediator—outcome associations or the interrelationship between the mediators. Mortality data were available till age 58 years and thus results apply to premature mortality; we are unable to infer whether associations will be stable through to later life. Finally, cause-specific mortality data was unavailable, restricting understanding of possible mechanisms linking ELAs to different causes of premature death.

Our main finding of varying associations for specific types of ELAs with risk of premature mortality is novel largely because there is a dearth of literature that focuses on such variations. The large population and range of ELA measures examined in our study compared with two previous studies[9 19] has facilitated this novel finding. Notably, in respect of CMs, we found that after accounting for confounders and all other ELAs, sexual and physical abuse and also neglect were independently associated with elevated mortality in mid-adulthood in the 1958 birth cohort, but there were no associations for psychological and witnessing abuse. The strong association for child sexual abuse, 2.6 times higher risk of premature mortality, is particularly important given the lack of evidence to date. One previous study of CM and mortality did not include sexual abuse[9] and a second study considered sexual abuse, physical abuse and neglect as a composite measure;[19] thus comparison with our findings for specific types of CMs is not possible. The latter composite measure study reported no association with mortality in young adulthood[19] whereas, our focus is on premature mortality for the age range 44/45–58 years. The life-stage examined might explain discrepant findings, that is, associations with mortality were not present in young adulthood[19] but may emerge by mid-adulthood as suggested here. For physical abuse, our finding of a 73% higher risk of premature mortality is consistent with a previous estimate of 58% higher risk of death for severe physical abuse in US women aged over 45–94 years at the end of follow-up.[9] This broad similarity in estimates for physical abuse was unexpected given the wider age range of US study participants compared with our age range of 44/45–58 years for mortality follow-up. Nonetheless, there was a discrepancy between our observed association for physical abuse and mortality and the lack of an association in US men.[9] The most common cause of death for men between 20 years and 49 years is due to external causes (eg, accidents and suicides), whereas, from age of 50 years, cancer, heart disease and strokes and respiratory diseases are the most common causes of death.[32] These variations in main cause of death may explain the noted discrepancy in findings. For child neglect we are unable to compare our finding of an independent association, with a 43% higher risk of premature death in mid-adulthood, as neither of the two previous CM—mortality studies investigated this exposure separately.[9 19] Thus, our finding provides new evidence for an important component of CM where knowledge of long-term outcome is particularly sparse.[30] For witnessing abuse in childhood, we are not aware of any previous study with which to confirm our null finding in relation to premature mortality; whereas for psychological abuse, findings for the USA (weak association in women only[9]) and our UK (null) study are discrepant. Possible reasons for discrepancies include differences in age at death, abuse measurement and also, the extent to which other ELAs were taken into account. In respect of the latter, it is noteworthy that our findings for specific CM associations with elevated mortality in mid-adulthood were independent of other types of CM as well as childhood socioeconomic circumstances, highlighting the potential for long-term harm associated with specific CMs.

A further novelty of our study is the demonstration that the early-life socioeconomic disadvantage association of an approximate doubling in risk of premature all-cause mortality was independent of specific CMs. While links between early-life socioeconomic disadvantage and mortality in adulthood are well established[2] and consistent with previous work in this cohort,[33] few studies[9] consider both CM and early-life socioeconomic disadvantage simultaneously. By suggesting that, notwithstanding the utility of understanding the long-term impact of CM, the latter does not appear to undermine or explain the strong and robust findings relating to childhood socioeconomic disadvantage our study adds new knowledge to the literature. This is important in a policy context as the recent emphasis on adverse childhood experiences may displace attention away from the early socioeconomic environment, as argued elsewhere.[13]

Our findings suggest that adult smoking may be a consistent and in some instances important explanatory factor across observed associations. This was expected because smoking remains one of the most common preventable causes of premature death in adults;[34] and, CM[6 21] and early-life socioeconomic disadvantage[35] are associated with subsequent smoking patterns. Thus, interventions to reduce smoking prevalence in specific ELA groups, either by reducing initiation or promoting cessation, might be considered as possible strategies to lessen differences in premature mortality. Interestingly, while specific CMs in this cohort were associated with the wide range of potential intermediary factors examined, these did not appear to explain associations with mortality. In particular, the strong association for sexual abuse was little explained by examined factors. Nonetheless, the potential intermediary factors considered here may play a role in pathways to mortality at older ages. Whereas in relation to the focus here on premature mortality, further insight into pathways from sexual abuse and other ELAs might be gained in future studies of cause-specific mortality.

In summary, our findings of independent associations for specific types of CM (sexual and physical abuse and neglect) and early-life socioeconomic disadvantage with increased risk of premature mortality in mid-adulthood highlight the long-lasting consequences of these ELAs. Smoking may be a particularly important intermediary for physical abuse, neglect and early-life socioeconomic disadvantage associations; adult socioeconomic factors may be an additional intermediary for neglect and early-life socioeconomic disadvantage. These findings are relevant for public health because, for example, an estimated 3.1 million adults in England and Wales reported being victims of sexual abuse before 16 years of age[36] and approximately 4.6 million children in the UK live in poverty.[4] Moreover, relative child poverty is projected to rise from 29.7% to 36.6% in the UK between 2018 and 2022.[37] Given these stark projections and our study findings of a strong relationship between childhood disadvantage and an early adult death, policies focused on improving socioeconomic opportunities and assistance to adopt and maintain positive health behaviours for individuals from disadvantaged backgrounds may reduce the burden of premature mortality.

## CONCLUSIONS

In sum, our findings highlight the potential of specific types of CMs (ie, sexual abuse, physical abuse and neglect) for long-term harm. Notwithstanding this important finding, childhood socioeconomic disadvantage associations with premature mortality are strong and not explained by associations with CM.

**Contributors** CP and SMPP conceived the study. NTR carried out the analysis and drafted the paper. All authors contributed to the interpretation of data, revision of the manuscript and approved its final version.

**Funding** Application for all-cause mortality was made through the UK Data service. This work was supported by US National Institute on Ageing (NIA) of the National Institutes of Health under award number U24AG047867 and the UK Economic and Social Research Council (ESRC) and the Biotechnology and Biological Sciences Research Council (BBSRC) and by the National Institute for Health Research Biomedical Research Centre at Great Ormond Street Hospital for Children NHS Foundation Trust and University College London. SMPP is funded by a UK Medical Research Council Career Development Award (ref: MR/P020372/1).

**Disclaimer** The views expressed in the publication are those of the authors and not necessarily those of the funders.

**Competing interests** None declared.

**Patient consent for publication** Not required.

**Ethics approval** Ethical approval was given at various sweeps, including at 50 years by the London Multicentre Research Ethics Committee. Participants gave informed consent at various sweeps.

**Provenance and peer review** Not commissioned; externally peer reviewed.

**Data availability statement** Data are available in a public, open access repository. Cohort data comply with ESRC data sharing policies, readers can access these data via the UK Data Archive at http://www.data-archive.ac.uk/. Applications for child abuse and all-cause mortality data is made through the UK Data service.

**ORCID iDs**
Nina T Rogers http://orcid.org/0000-0003-1857-2122
Snehal M Pinto Pereira http://orcid.org/0000-0002-0876-8757

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
