## [Reviewer comments · BMJ Open]

ARTICLE DETAILS

TITLE (PROVISIONAL)	Child maltreatment, early life socioeconomic disadvantage and all-cause mortality in mid-adulthood: findings from a prospective British birth cohort
AUTHORS	Rogers, Nina; Power, Christine; Pinto Pereira, Snehal

VERSION 1 – REVIEW

REVIEWER	Bridger, Emma Birmingham City University
REVIEW RETURNED	10-May-2021

GENERAL COMMENTS	Many thanks for inviting me to review this manuscript. I found it very well written and enjoyable to read. I agree with the authors about the value and importance of this work and found the rationale and conclusions to be well supported by the data and analyses presented. The key findings reported are the distinct associations between different kinds of early life adversities (when controlling for other adversities) and mortality between 44-58 years of age. Particularly interesting is the comparative impact of the distinct intermediary pathways on the specific ELA-mortality associations, which highlights a notable role for smoking in this cohort and comparatively minor role for adult socioeconomic factors. I also agree with the authors that a key novelty of this piece of work is in demonstrating an association between early life socioeconomic disadvantage with mortality when controlling for other ELAs which are likely to be clustered with disadvantage. Although the 1958 cohort is a specific cohort which brings its own strengths and weaknesses (which the authors are very aware of), this is nonetheless a potent finding, and the authors do well to identify this and highlight the policy implications. I have the following mainly minor comments, most of which relate to further detail and/or justification for particular reporting/analytic decisions. • Table 3: Does socioeconomic disadvantage in models 2 (and 3) no longer predict mortality when the full list of adjustments is included? Can the authors speak more to this decision to adjust for this only for the non-socioeconomic disadvantage ELAs? Relatedly, can you confirm/clarify that the inclusion of social class at birth (or age 7) for the other ELAs in models 2 and 3 is a proxy for adjusting for socioeconomic disadvantage in model 2?
---

	 • Neglect data was considered non-missing if data was available for at least 6 out of the 11 items. Can the authors indicate why 6 was selected as the cut-off and whether this approach was employed in previous work used to validate this measure of neglect (Power et al., 2020)? • Page 8: Imputed results are reported because these were deemed to be 'similar to those obtained using observed values'. On what basis were these deemed sufficiently similar? • At the top of page 8, the weak to moderate correlations between ELAs is referred to. I think readers would like to see a table of the correlations in the supplementary tables – I know I would like to see this. • It might help some readers to make explicit that analysis of intermediary mechanisms was limited to those ELAs which were found to predict mortality after adjustment of all other ELAs, and that this is why witnessing abuse and psychological abuse are absent from Table 4. • The term “mid-adult” or “mid-adulthood” is used to refer to a range of ages in the current manuscript. This is most notable perhaps for reference to the adult SES measures, which were actually taken at age 33, and therefore refer to an earlier time in the lifespan. It would be helpful to make the time point at which adult-intermediary data was collected more salient in the appropriate methodological section, rather than referring more generally to adulthood or mid-adulthood. • How do the authors account for the increase in HRs (Table 4) for physical and sexual abuse, once adiposity and cardio-metabolic factors are accounted for? Was this expected? The comparable impact on the ELA-mortality association in Table 4 is interesting given that the cardio-metabolic data appears to have no robust association with mortality in the current sample (Table S4). • Page 9: The sentence beginning “Whilst for early-life socioeconomic disadvantage” is incomplete/ungrammatical. • Page 11: I did not understand what the authors were referring to on line 29 when they state “possibly due to known variations in main causes of death by age and gender”. What does this refer to and how does this reconcile the current discrepancy? • On a related point, gendered effects are referred to but not actually reported in the current manuscript. Interaction sex effects are referred to on page 7 however, it is not explicit what sex/gender was interacted with here. Given the comparison with other work reporting gendered effects in the discussion, for the sake of completeness it would be useful to see such interactions in supplementary tables. • There are some formatting (spacing/comma) inconsistencies in Table S3.
--	---

REVIEWER	Soares, Ana Luiza University of Bristol
REVIEW RETURNED	15-Jun-2021

GENERAL COMMENTS	bmjopen-2021-050914 - Child maltreatment, early life socioeconomic disadvantage and all-cause mortality in mid-adulthood: findings from a prospective British birth cohort
--

The authors assessed the association of different types of child maltreatment and socioeconomic disadvantage with premature death in 9,310 middle-aged adults from a British birth cohort, and explored potential intermediate pathways of these associations. The topic is of great relevance and the work is novel with potential implications for further research in the area. My comments regarding the rigor of the study and suggestions for improvement of the manuscript are listed below.

Major comments:

- The suggestion of different pathways by different types of ELAs is very interesting. However, more caution is needed in the interpretation of the results. With the analysis performed, it is not possible to affirm that smoking may be a particularly important factor. It is possible that this pathway is confounded by other factors, such as adult socioeconomic position, mental health and other behavioural factors. The analysis performed is important to give insights of possible mechanisms, but trying to determine the most important component is not possible without considering confounding between mediator-outcome association and the interrelationship between the mediators. As suggested recently using data from UK Biobank (BMC Med 2020; 18(143); Int J Epidemiol 2021; dyab085) even though smoking plays an important role in the association between childhood maltreatment and cardiovascular disease, mental health seems to be a relatively more important factor.

- It is not clear why child-to-adult growth was assessed as a possible intermediate. It is not associated with mortality (Table S1) and it is not well justified in the manuscript why to include this specific factor.

- As described in the second paragraph of the statistical analysis, possible sex differences in the associations have been explored. Even though differences by sex are present in the associations between early life adversities (ELAs) and many health outcomes, and have also been mentioned in the introduction, this was not specified as an objective in the manuscript. I suggest to include that.

- When describing the adjustment for potential intermediates, it is not clear in the methods that these were assessed in groups as well as each factor separately. That only becomes clear after seeing the results.

- Multiple imputation has been used for missing data. However, it is not clear what data have been imputed. Although the authors say that imputed values were similar to observed values, it would be helpful to have a supplementary table with the proportion of missing data for each variable, as well as distribution of observed and imputed values.

- That said, it is unclear who was included in the analysis. By reading the description of the first sensitivity analysis in the methods, it seems that the main analysis was restricted to those who completed the child maltreatment questions at 44-45y. Does this include those who responded at least one question? The numbers (presented in Table 2) vary for the different

types of ELA (including those reported at 44-45y). Complete data seems to be available only for psychological abuse; have the other exposures been imputed?

- It would be important to have a supplementary table with descriptive statistics of the intermediate factors. In Table S1 footnote it is mentioned that for categorical variables extreme category groups were compared, but it is not possible to know what is the proportion in each group.
- I believe that linkage to mortality records was available until end of 2016 at the time of analysis. Given premature mortality is usually measured in terms of potential years of life lost (PYLL) before the age of 70 years, and the current study assessed up to 58 years, it would be important to highlight the fact that linkage was available until 2016 as a limitation. Some of the independent associations could have remained evident (e.g. witnessing abuse) with a higher number of cases, and therefore a narrower confidence interval.

Minor comments:

- In the conclusion of the abstract, the authors could add that different types of ELAs might affect premature mortality via different pathways and that this requires further exploration. If space is a limitation, the results section could be simplified.
- The authors should acknowledge previous work carried out in this cohort assessing the associations between adverse childhood experiences and premature all-cause mortality (Eur J Epidemiol (2013) 28:721–734)
- Limitations: child maltreatment comprised the period until 18 years, and data for neglect is available until 11 years of age, therefore the prevalence of neglect might be underestimated. This should be mentioned in the limitations.
- I suggest to present the results from the sensitivity analyses performed, currently in the methods section, in the results section.
- Table 2: one decimal point is enough for the prevalence
- Results from table 3 could be presented as a figure.
- Avoid repetition of the results in the discussion (e.g. [...] “Whilst for early-life socioeconomic disadvantage, an approximate doubling in risk of premature mortality applied to one in 10 of the population. For physical abuse and neglect the estimated elevated risk of death was more modest (73% and 43% higher respectively), whereas no associations were observed for psychological and witnessing abuse”).
- In line 6, page 11, the authors say “... there was no associations for psychological and witnessing abuse”; I would suggest complementing that sentence with independently of the

	other ELAs, as these were associated with mortality when adjusted for confounders (model 2).
--	--

VERSION 1 – AUTHOR RESPONSE

Reviewer: 1

Response: thank you for acknowledging the value and importance of our work and noting that the rationale, conclusions, and highlighted policy implications are well supported by the data and analyses presented.

Response: Figure 1 (which replaces Table 3 at the request of Reviewer 2) shows that socioeconomic disadvantage in models 2 and 3 predicts mortality. We decided that the association for socioeconomic disadvantage should not be adjusted for other socioeconomic markers (e.g. overcrowding) to prevent over-adjustment, whilst establishing the independence of disadvantage from other ELAs (as was an aim of the study justifying model 3 in our analysis). To clarify this latter point we have inserted the word ‘independently’ in the aims as set out in the last paragraph of the introduction. Details that address the query on inclusion of factors in models 2 and 3 are shown in Figure 1 footnotes. As the footnotes indicate, when socioeconomic disadvantage was our exposure of interest the covariates included in model 2 were: sex, maternal age at birth, birthweight, birth order and 7y impairment. For neglect and the different types of abuse, we additionally adjusted model 2 (and model 3) for social class at birth, 7y household amenities, 7y housing tenure, and 7y household crowding. We confirm that social class at birth is included as a proxy for socioeconomic disadvantage (defined in table 1).

Neglect data was considered non-missing if data was available for at least 6 out of the 11 items. Can the authors indicate why 6 was selected as the cut-off and whether this approach was employed in previous work used to validate this measure of neglect (Power et al., 2020)?

Response: Our decision to use a cut-off of 6 out of 11 items is a pragmatic one. 65% of the sample had data on all 11 items; 91% had data on at least 6 of the 11 items. Our cut-off ensures that over 50% of the individual items are non-missing and for those missing more than 6 items, we treat the variable neglect as unobserved (and impute using multiple imputation). This approach has been extensively used in previous work (Power et al. 2020) – which we now reference in the footnotes to Table 2.

Page 8: Imputed results are reported because these were deemed to be ‘similar to those obtained using observed values’. On what basis were these deemed sufficiently similar?

Response: To support the statement ‘Imputed results were similar to those obtained using observed values...’ we now provide in the supplementary section (Table S6) a comparison of the imputed and complete case analysis. In addition, at the request of reviewer 2, we now also provide details of the proportion of missing data in our analytic sample and the distribution of observed and imputed analysis samples in Table S5.

At the top of page 8, the weak to moderate correlations between ELAs is referred to. I think readers would like to see a table of the correlations in the supplementary tables – I know I would like to see this.

Response: We now provide these data in the supplementary section (Table S4).

It might help some readers to make explicit that analysis of intermediary mechanisms was limited to those ELAs which were found to predict mortality after adjustment of all other ELAs, and that this is why witnessing abuse and psychological abuse are absent from Table 4.

Response: We now make this point explicitly at the end of the 2nd paragraph under “Statistical analysis”.

The term “mid-adult” or “mid-adulthood” is used to refer to a range of ages in the current manuscript. This is most notable perhaps for reference to the adult SES measures, which were actually taken at age 33, and therefore refer to an earlier time in the lifespan. It would be helpful to make the time point at which adult-intermediary data was collected more salient in the appropriate methodological section, rather than referring more generally to adulthood or mid-adulthood.

Response: We have added in the specific ages at which data were collected for the adult-intermediaries in the section “Potential mid-adult intermediary factors”. In addition, we now explicitly state the ages in relevant tables (Tables S1, S2, S5 and S7).

How do the authors account for the increase in HRs (Table 4) for physical and sexual abuse, once adiposity and cardio-metabolic factors are accounted for? Was this expected? The comparable impact on the ELA-mortality association in Table 4 is interesting given that the cardio-metabolic data appears to have no robust association with mortality in the current sample (Table S4).

Response: We agree that the point estimates for associations for sexual abuse, in particular, increase slightly after accounting for either adiposity or cardio-metabolic factors. In response, we offer two points for consideration. First, the changes (both increases and decreases) in point estimates need to be interpreted in the context of their confidence intervals. Overlap in CIs was the justification behind our conclusion in the discussion that “... in relation to potential intermediaries...., in some instances this attenuation was minor, such that for sexual abuse the association was largely unaltered.” For example, this applies to the HR for sexual abuse (model 3) before accounting for adiposity (2.64(1.52,4.59)) vs after accounting for adiposity (2.71(1.56,4.73)), i.e. the effect of adjustment is weak. Second, if the point estimates are detecting a real increase in HR, this implies that risks for mortality associated with sexual abuse are greater once account is taken of adiposity and cardio-metabolic factors. However, because of the first main point mentioned above, we have made no further additions to the text given that we already refer to minor effects of adjustment.

Finally, it may seem surprising that the cardiometabolic data have no robust association in the current sample but note that the outcome is premature mortality (44/45y to 58y) rather than mortality at later ages (where we might expect stronger associations). We now include edits (mentioned below) on differences in main causes of death at different life-stages. These edits, pointing to external causes as a common cause of death for men aged 20-49y (whereas cancer, heart disease, strokes and respiratory causes are more common after age 50y) provide readers with an explanation for cardio-metabolic factor associations in tables 4 and S2.

Page 9: The sentence beginning “Whilst for early-life socioeconomic disadvantage” is incomplete/ungrammatical.

Response: We have edited this sentence for clarity.

Page 11: I did not understand what the authors were referring to on line 29 when they state “possibly due to known variations in main causes of death by age and gender”. What does this refer to and how does this reconcile the current discrepancy?

Response: We have edited the discussion to clarify why there may be discrepancies between our finding of an association for physical abuse and mortality, whilst a previous study in the US found no association in men. Discrepancies in findings could be due to the reasons for mortality in the different age ranges considered (mortality follow-up was age 44/45y to 58y in our study versus aged over 45y to 94y at the end of follow-up in the US study). For men aged 20 to 49y, the most common cause of death is due to external causes (e.g. accidents and suicides). From age 50y, cancer, heart disease and strokes and respiratory diseases were the most common causes of death. Therefore, the cause of death in men in our sample and the US sample is likely to vary due to differences in the age-range examined. This might explain differences in the association between physical abuse and mortality in the two study populations.

On a related point, gendered effects are referred to but not actually reported in the current manuscript. Interaction sex effects are referred to on page 7 however, it is not explicit what sex/gender was interacted with here. Given the comparison with other work reporting gendered effects in the discussion, for the sake of completeness it would be useful to see such interactions in supplementary tables.

Response: We have edited the second paragraph of the statistical analysis to clarify that the interactions tested were for the association of each type of ELA and mortality by sex. As requested, we also now provide results of sex-specific analysis in Table S3.

There are some formatting (spacing/comma) inconsistencies in Table S3.

Response: We have corrected the formatting of (what is now) Table S9.

Reviewer: 2

The authors assessed the association of different types of child maltreatment and socioeconomic disadvantage with premature death in 9,310 middle-aged adults from a British birth cohort, and explored potential intermediate pathways of these associations. The topic is of great relevance and the work is novel with potential implications for further research in the area. My comments regarding the rigor of the study and suggestions for improvement of the manuscript are listed below.

Response: Thank you for acknowledging the relevance and novelty of our work.

Major comments:

- The suggestion of different pathways by different types of ELAs is very interesting. However, more caution is needed in the interpretation of the results. With the analysis performed, it is not possible to affirm that smoking may be a particularly important factor. It is possible that this pathway is confounded by other factors, such as adult socioeconomic position, mental health and other behavioural factors. The analysis performed is important to give insights of possible mechanisms, but trying to determine the most important component is not possible without considering confounding between mediator-outcome association and the interrelationship between the mediators. As suggested recently using data from UK Biobank (BMC Med 2020; 18(143); Int J Epidemiol 2021; dyab085) even though smoking plays an important role in the association between childhood maltreatment and cardiovascular disease, mental health seems to be a relatively more important factor.

Response: We agree that with the analysis performed, it is not possible to affirm that smoking is an important factor but that it provides insights of possible mechanisms. We have revised the discussion (2nd para) to include the following addition: 'The analytic approach used allows insights into possible mechanisms underlying ELA associations with premature mortality, but it is not possible to determine the most important intermediaries without considering confounding between mediator-outcome associations or the interrelationship between the mediators.' We have also checked that our discussion of potential mediating factors uses suggestive rather than deterministic language.

- It is not clear why child-to-adult growth was assessed as a possible intermediate. It is not associated with mortality (Table S1) and it is not well justified in the manuscript why to include this specific factor.

Response: Thank you for pointing this out. We have now removed reference to child-to-adult growth as a possible intermediate from the manuscript.

- As described in the second paragraph of the statistical analysis, possible sex differences in the associations have been explored. Even though differences by sex are present in the associations between early life adversities (ELAs) and many health outcomes, and have also been mentioned in the introduction, this was not specified as an objective in the manuscript. I suggest to include that.

Response: As explained in the introduction, the main focus (and novelty) of our paper is on type of child maltreatment and mortality as we are aware of only two previous studies on this topic. With only one of

these studies reporting differences by sex, the rationale for a main focus on sex differences was not compelling. Whilst not a main aim here, it was important to test whether a combined analysis of men and women was justified, hence the tests for interactions by sex to which the reviewer refers. However, in view of reviewer's comments we now include edits on sex differences in the second paragraph under 'Statistical analysis' (see response to Reviewer 1 above). In particular, with scarce evidence available, the addition of Table S3 provides information on (lack of) differences by sex in the association of type of ELA and mortality.

- When describing the adjustment for potential intermediates, it is not clear in the methods that these were assessed in groups as well as each factor separately. That only becomes clear after seeing the results.

Response: We have edited the 2nd paragraph of the statistical analysis to explicitly state that we considered the potential intermediaries in groups as well as for each factor separately.

- Multiple imputation has been used for missing data. However, it is not clear what data have been imputed. Although the authors say that imputed values were similar to observed values, it would be helpful to have a supplementary table with the proportion of missing data for each variable, as well as distribution of observed and imputed values.

Response: We now also provide details of the proportion of missing data in our analytic sample and the distribution of observed and imputed analysis samples in Table S5. In addition, we now also provide, in Table S6, a comparison of the imputed and complete case analysis.

- That said, it is unclear who was included in the analysis. By reading the description of the first sensitivity analysis in the methods, it seems that the main analysis was restricted to those who completed the child maltreatment questions at 44-45y. Does this include those who responded at least one question? The numbers (presented in Table 2) vary for the different types of ELA (including those reported at 44-45y). Complete data seems to be available only for psychological abuse; have the other exposures been imputed?

Response: We have edited the first paragraph of the methods to state that the sample consisted of those who completed at least one question on maltreatment. By providing Table S5 (at the reviewer's recommendation above), we now demonstrate the (i) range of missing data in our dataset: 0.02% (for physical, sexual and witnessing abuse) to 21% (LDL-c) and (ii) substantive model variables (i.e. variables that are included in models 1,2 and/or 3) that are included in the imputation models. We have edited the last paragraph in the 'Statistical analysis' section to provide more explicit details of missing data in the sample and the variables included in the imputation models.

- It would be important to have a supplementary table with descriptive statistics of the intermediate factors. In Table S1 footnote it is mentioned that for categorical variables extreme category groups were compared, but it is not possible to know what is the proportion in each group.

Response: We now provide this information in (a new) Table S1.

- I believe that linkage to mortality records was available until end of 2016 at the time of analysis. Given premature mortality is usually measured in terms of potential years of life lost (PYLL) before the age of 70 years, and the current study assessed up to 58 years, it would be important to highlight the fact that linkage was available until 2016 as a limitation. Some of the independent associations could have remained evident (e.g. witnessing abuse) with a higher number of cases, and therefore a narrower confidence interval.

Response: In the second paragraph of the discussion, we now acknowledge that mortality data was available till age 58y and we are unable to infer whether associations will be stable through to later life. See all response to reviewer 1 above, regarding differences in main causes of death by age.

Minor comments:

- In the conclusion of the abstract, the authors could add that different types of ELAs might affect premature mortality via different pathways and that this requires further exploration. If space is a limitation, the results section could be simplified.

Response: We have edited the abstract conclusions to include this point.

- The authors should acknowledge previous work carried out in this cohort assessing the associations between adverse childhood experiences and premature all-cause mortality (Eur J Epidemiol (2013) 28:721–734)

Response: We now acknowledge this previous work in the first paragraph of the introduction.

- Limitations: child maltreatment comprised the period until 18 years, and data for neglect is available until 11 years of age, therefore the prevalence of neglect might be underestimated. This should be mentioned in the limitations.

Response: We now acknowledge this limitation in the second paragraph of the discussion.

- I suggest to present the results from the sensitivity analyses performed, currently in the methods section, in the results section.

Response: We have removed the results of the sensitivity analyses from the methods and present them at the end of the results section.

- Table 2: one decimal point is enough for the prevalence

Response: We have edited the table to give one decimal point.

- Results from table 3 could be presented as a figure.

Response: We now provide a figure instead of (old) table 3.

- Avoid repetition of the results in the discussion (e.g. [...] “Whilst for early-life socioeconomic disadvantage, an approximate doubling in risk of premature mortality applied to one in 10 of the population. For physical abuse and neglect the estimated elevated risk of death was more modest (73% and 43% higher respectively), whereas no associations were observed for psychological and witnessing abuse”).

Response: We appreciate that it may seem a little repetitive but, it is our view that at this point in the paper it is important to recap main findings before elaborating in the discussion.

- In line 6, page 11, the authors say “... there was no associations for psychological and witnessing abuse”; I would suggest complementing that sentence with independently of the other ELAs, as these were associated with mortality when adjusted for confounders (model 2).

Response: We have edited the sentence to clarify that we are referring to model 3 (i.e. after accounting for confounders and all other ELAs).

VERSION 2 – REVIEW

REVIEWER	Soares, Ana Luiza University of Bristol
REVIEW RETURNED	03-Aug-2021
GENERAL COMMENTS	The authors did a great job revising the manuscript. I am satisfied with their responses to my comments.